# Polyarginine Cell-Penetrating Peptides Bind and Inhibit SERCA2

**DOI:** 10.3390/cells12192358

**Published:** 2023-09-26

**Authors:** Per Kristian Lunde, Ornella Manfra, Thea Parsberg Støle, Marianne Lunde, Marita Martinsen, Cathrine Rein Carlson, William E. Louch

**Affiliations:** Institute for Experimental Medical Research, Oslo University Hospital and University of Oslo, 0450 Oslo, Norway; p.k.lunde@medisin.uio.no (P.K.L.); ornella.manfra@medisin.uio.no (O.M.); t.p.stole@medisin.uio.no (T.P.S.); marlu@medisin.uio.no (M.L.); marita.martinsen@medisin.uio.no (M.M.); w.e.louch@medisin.uio.no (W.E.L.)

**Keywords:** cell-permeating peptides, calcium homeostasis, SERCA2

## Abstract

Cell-penetrating peptides (CPPs) are short peptide sequences that have the ability to cross the cell membrane and deliver cargo. Although it is critical that CPPs accomplish this task with minimal off-target effects, such actions have in many cases not been robustly screened. We presently investigated whether the commonly used CPPs TAT and the polyarginines Arg_9_ and Arg_11_ exert off-target effects on cellular Ca^2+^ homeostasis. In experiments employing myocytes and homogenates from the cardiac left ventricle or soleus muscle, we observed marked inhibition of Ca^2+^ recycling into the sarcoplasmic reticulum (SR) following incubation with polyarginine CPPs. In both tissues, the rate of SR Ca^2+^ leak remained unchanged, indicating that protracted Ca^2+^ removal from the cytosol stemmed from inhibition of the SR Ca^2+^ ATPase 2 (SERCA2). No such inhibition occurred following treatment with TAT, or in preparations from the SERCA1-expressing extensor digitorum longus muscle. Experiments in HEK cells overexpressing individual SERCA isoforms confirmed that polyarginine incubation specifically inhibited the activity of SERCA2a and 2b, but not SERCA1 or 3. The attenuation of SERCA2 activity was not dependent on the presence of phospholamban, and ELISA-based analyses rather revealed direct interaction between the polyarginines and the actuator domain of the protein. Surface plasmon resonance experiments confirmed strong binding within this region of SERCA2, and slow dissociation between the two species. Based on these observations, we urge caution when employing polyarginine CPPs. Indeed, as SERCA2 is expressed in diverse cell types, the wide-ranging consequences of SERCA2 binding and inhibition should be anticipated in both experimental and therapeutic settings.

## 1. Introduction

Cell-penetrating peptides (CPPs) are short peptide sequences of 5 to 30 amino acids, derived from natural or synthetic sources, that have the ability to cross the cell membrane and deliver cargo [1]. The delivered cargo molecules may include proteins, nucleic acids, or small molecules. Indeed, CPPs have been shown to provide a versatile and efficient means of delivering various bioactive molecules into cells, including drugs, gene therapies, and imaging agents. For this reason, CPPs have been extensively studied in biomedical research, and continue to be explored for therapeutic applications such as intracellular targeting, gene editing, and diagnostics [2].

The cell membrane is a selective barrier that prevents the entry of large and charged molecules. However, CPPs possess unique properties that allow them to overcome this barrier. Many CPPs exhibit highly hydrophilic and cationic properties that allow them to interact with anionic components of the phospholipid bilayer [3]. Ensuing alterations in the membrane structure are believed to enable endocytosis and/or translocation of the peptide, depending on the peptide’s structure, concentration, and cargo [4]. For example, peptides containing arginine repeats (i.e., polyarginines) or transactivator of transcription (TAT) sequences, have been shown to bind to phospholipid phosphate groups, which destabilizes the membrane and creates a pore that allows peptide entry [5,6]. Cellular entry by this mechanism has been shown to be highly efficient [7].

While CPPs have generally been shown to exhibit low cytotoxicity and high cell penetration efficiency, the design and optimization of CPPs for specific applications nevertheless requires the careful consideration of factors such as peptide sequence, cargo type, and cellular targeting to ensure the desired outcomes and minimize potential off-target effects. Given our own interest in intracellular Ca^2+^ homeostasis, we presently examined whether the commonly used CPPs Arg_9_, Arg_11_, and TAT have off-target effects on intracellular Ca^2+^ uptake and release. Our results demonstrate that polyarginine CPPs bind to and inhibit the (sarco)endoplasmic reticulum (SR) Ca^2+^ ATPase type-2 (SERCA2), which complicates their pertinence for both experimental and therapeutic applications.

## 2. Materials and Methods

### 2.1. Ethical Approval

All protocols were approved by the Norwegian Animal Research Authority and performed in accordance with the Norwegian Animal Welfare Act and NIH Guidelines. Experiments employed tissue and cells isolated from adult male Wistar rats (250–350 g), wild-type C57BL/6NT mice (20–30 g), and phospholamban knockout mice (20–30 g) [8]. Animals were housed with ad libitum access to food and water, and maintained in a temperature- and humidity-controlled facility, on a 12 h:12 h light–dark cycle.

### 2.2. Peptides and Recombinant Protein

The following peptides were synthesized with or without an *N*-terminal biotin tag at >80% purity by GenScript Biotech Corp (Rijswijk, The Netherlands) for experimental testing:TAT: RKKRRQRRRArg_9_: RRRRRRRRRArg_11_: RRRRRRRRRRRCargo: IEKELAQQYQNADAITLE, a scrambled control sequence of RIAD (LEQYNQLADQIIKEATEK) which is a high-affinity disruptor peptide for protein kinase A (PKA) type I [9].Cargo_1–9_: IEKELAQQYCargo_8–18_: QYQNADAITLECargo_1–11_: IEKELAQQYQNRecombinant 12xHis-SERCA2_111–253_, containing the actuator domain, also known as the A-domain, of SERCA2 [10], and His-CaMKIIδ_1–165_ were generated by Genscript.

### 2.3. Isolated Cardiomyocyte Experiments

For single cell experiments, rat left ventricular cardiomyocytes were isolated as described previously [11,12]. In brief, rats were anaesthetized by inhalation of a mixture of 5% isoflurane and 95% O_2_. Hearts were then rapidly excised and mounted via the aorta on a Langendorff apparatus to enable retrograde perfusion with buffer containing collagenase Type II (Worthington Biochemical Corporation, Lakewood, NJ, USA). Following digestion, the heart was cut down, diced, and agitated, and isolated cardiomyocytes were then filtered, washed, and sedimented. Cells were stored in a solution containing (in mmol/L) 130 NaCl, 5.4 KCl, 25 HEPES, 0.5 MgCl_2_, 0.4 NaH_2_PO_4_, 22 glucose, 0.2 CaCl_2_ (pH 7.40, 22 °C) until use, within 6 h of isolation.

Prior to experiments, cardiomyocytes were preincubated for 1 h with 5 µmol/L of the individual CPPs added to the cell storage solution. During the last 10 min of incubation, cells were additionally loaded with fluo-4 AM (5 µmol/L, Thermo Fisher Scientific, Waltham, MA, USA) for Ca^2+^ imaging. The cells were then plated on the stage of an inverted microscope and perfused with HEPES-Tyrode solution containing (in mmol/L) 140 NaCl, 0.5 MgCl_2_, 5.0 HEPES, 5.5 glucose, 0.4 NaH_2_PO_4_, 5.4 KCl, and 1.8 CaCl_2_ at pH 7.4, 37 °C. Whole-cell Ca^2+^ transients were recorded by wide-field photometry, as described previously [13], using a D-104 microscope photometer and photomultiplier tube (Horiba Ltd., Kyoto, Japan). During field stimulation at 1 Hz, fluorescence was excited by a 475 nm light-emitting diode, and emission was collected at >510 nm.

Ca^2+^ signals were analysed by first subtracting background fluorescence values from the cellular recordings. Ca^2+^ transient magnitude was calculated as peak fluorescence divided by resting fluorescence (F/F_0_). The decay time of Ca^2+^ transients was assessed using a monoexponential fit of the declining phase of the Ca^2+^ transient to calculate tau (s), using Clampfit 10.4 software (Molecular Devices LLC, Sunnyvale, CA, USA).

### 2.4. Ca^2+^ Homeostasis in Ventricular and Muscle Homogenates

Ca^2+^ handling was also examined in homogenates prepared from mouse left ventricle, rat soleus, and rat extensor digitorum longus (EDL) muscles using methods described by O’Brien [14] and modified by Li et al. [15]. Freshly excised left ventricles and muscles were weighed and homogenized in ice cold homogenization buffer (1:10 wet weight/vol, pH 7.9) containing (in mmol/L) 300 sucrose, 5 NaN_3_, 1 EDTA, 40 L-histidine, 40 Tris HCl, and protease inhibitors (Complete EDTA-free No 05056489001, Roche, Mannheim, Germany). Homogenization was performed with a Polytron 1200 (Kinematica AG, Luzern, Switzerland) at 25,000 rpm for 3 × 20 s, with a 20 s break between bursts. Homogenates were then aliquoted, frozen in liquid N_2_, and stored at −80 °C until use.

10–20 µL of freshly thawed and vortexed homogenate was added to 250 µL of “assay” buffer, containing (in mmol/L) 165 KCl, 22 Hepes, 7.5 oxalate, 11 NaN_3_, 0.0055 TPEN, 4.5 MgCl_2_, 9 Tris HCl, and 0.002 Fura-2 salt (pH 7.0, 37 °C). Each CPP was added to attain a final concentration of 40 µmol/L and incubated for 5 min at 37 °C prior to experiments.

Ca^2+^ fluxes were monitored as Fura-2 fluorescence ratio (340 nm/390 nm) using a Hidex Sense Multimodal Microplate Reader (Kem-En-Tec Nordic AS, Uppsala, Sweden). Ca^2+^ uptake by the vesicles was initiated by the addition of Na_4_ATP (2.2 mmol/L) and blocked by the application of thapsigargin (1.5 µmol/L) after 12 min [16]. SR Ca^2+^ content, i.e., the “releasable Ca^2+^” was estimated by adding the ryanodine receptor opener 4-Chloro-m-cresol (CmC, 5.5 mmol/L) [17]. The fluorescence ratio was calibrated to [Ca^2+^] using the following equation: [Ca^2+^] = K_d_ × ((R − R_min_)/(R_max_ − R)) × (S_f2_/S_b2_), where R is the 340 nm/390 nm fluorescence ratio, and K_d_ is the dissociation constant of Fura-2 and Ca^2+^ (224 nmol/L) [18,19]. R_min_ is the fluorescence ratio at very low [Ca^2+^] and R_max_ is the ratio at saturating [Ca^2+^], obtained by sequentially adding 3.3 mmol/L EGTA and 4.8 mmol/L CaCl_2_ to the well at the end of each recording. S_f2_/S_b2_ is the ratio of measured fluorescence intensity at 390 nm when Fura-2 is Ca^2+^-free (3.3 mmol/L EGTA) or Ca^2+^-bound (4.8 mmol/L CaCl_2_).

Ca^2+^ recordings were smoothed using TableCurve 2D (V 5.01, Systat Software Inc., Palo Alto, CA, USA) prior to analysis. After the addition of Na_4_ATP, the rate constant for Ca^2+^ uptake (k) was calculated by fitting the uptake curve to Ca^2+^ = Ca^2+^_0_ + ae^−t×k^ (Sigmaplot V14.0, Systat Software Inc.). Ca^2+^ leak from the vesicles was calculated as the linear rise (slope) of the Ca^2+^ signal after blockade of SERCA by thapsigargin. Leak measurements were normalized to SR content, determined as the peak [Ca^2+^] after addition of CmC.

### 2.5. Cell Culture and Transfection of HEK293 Cells

Human embryonic kidney (HEK) 293 cells were cultured to confluence in 10 cm Petri dishes in DMEM supplemented with 10% foetal calf serum, 100 U/mL penicillin, and 1% nonessential amino acids [20]. Cells were maintained at 37 °C and 5% CO_2_ in a humidified incubator, and transfected using Lipofectamine 2000, as described by the manufacturer (Invitrogen Dynal, Oslo, Norway). The employed gene constructs included rat *SERCA1* (NM_058213.1), rat *SERCA2a* (NP_001103609.1), rat *SERCA2b* (NP_001104293.1), and rat *SERCA3* (NM_012914.1) with *GFP* present at the N-terminus (pEGFP-C2 vector) (custom-made by GenScript). Following 24 h transfection, cells were washed with 5 mL 0.9% NaCl before harvesting in 1 mL homogenization buffer (see above in Section 2.4). The cell suspension was homogenized as described for ventricular and muscle tissue, and then aliquoted, frozen in liquid N_2_, and stored at −80 °C until use.

### 2.6. ELISA Analyses

A 96-well ELISA plate was coated with 1 × PBS (control), recombinant 12xHis-SERCA2_111–253_ (containing the A-domain, 1 µg/well) or His-CaMKIIδ_1–165_ (amino acids 1–165, 1 µg/well) (control) per well and incubated overnight at 4 °C with gentle agitation. Thereafter, wells were washed once in PBS-T (0.1% Tween-20, #1610781, Bio-Rad, Hercules, CA, USA) before blocking with 0.5% gelatine (G-1890, Sigma Merck, St. Louis, MO, USA) for one hour at room temperature. The coated wells were then incubated for 2 h at 37 °C with 50 µmol/L of biotin-labelled control peptides or CPPs (biotin-TAT, biotin-Arg_9_, or biotin-Arg_11_) with gentle agitation. The control peptides consisted of 9–11 amino acids (biotin-cargo_1–9,_ biotin-cargo_8–18_, or biotin-cargo_1–11_), derived from a scrambled control peptide sequence of 18 amino acids [9]. The wells were washed five times with PBS-T before being incubated with a monoclonal anti-biotin-HRP antibody (A0185, Sigma Merck, Darmstadt, Germany) for 30 min at room temperature. Five additional washes with PBS-T were then performed, before the wells were incubated with 100 µL of Ultra TMB substrate solution (34028, Thermo Fisher Scientific, Waltham, MA, USA) for 15–30 min at room temperature with gentle agitation (blue colour indicated interaction). The reaction was stopped with 100 µL 2 mol/L hydrochloric acid (the blue colour changed to yellow), and the absorbance of each well was measured at 450 nm (Hidex Sense multimodal microplate reader, Kem-En-Tec Nordic AS, Turku, Finland).

### 2.7. Surface Plasmon Resonance

A Biacore X100 (Biacore Inc., Uppsala, Sweden) was employed for surface plasmon resonance analyses. Streptavidin (SA) chips (BR100032, Cytiva, Marlborough, MA, USA) were conditioned with three consecutive 1 min injections of 1 × Biacore running buffer (BR100826) before biotinylated Arg_11_ was immobilized at 744.3 resonance units (RUs). Recombinant 12xHis-SERCA2_111–253_ [21], containing the SERCA2 A-domain, was dialysed into 1 × Biacore running buffer before increasing concentrations of 12xHis-SERCA2_111–253_ were injected over the sensor surface at a flow rate of 30 µL/minute for 180 s. For the 3 experimental runs performed, the employed concentration range of 12xHis-SERCA2_111–253_ was 0.7–200 nM, 0.19–50 nM, or 12.5–200 nM. To be able to measure the slow dissociation rate of the interaction, the dissociation time of the instrument was set to 600 or 1000 s. Sensorgrams were analysed using Biacore X100 evaluation software with the assumption of one-to-one binding (Langmuir binding model).

### 2.8. Statistics

Data are presented as mean ± standard error of the mean. All data were tested for normality of distribution using the Shapiro–Wilk test. Normally distributed data were statistically tested with a one-way ANOVA with Tukey’s post hoc correction for multiple comparisons. Non-normal distributions were examined by Kruskal–Wallis analysis on ranks with Dunn’s post hoc. A *p*-value < 0.05 was considered statistically significant. All statistical analyses were performed using GraphPad Prism 9.4.1 (La Jolla, CA, USA) or Sigmaplot v14.0, Systat Software Inc. (Palo Alto, Ca, USA).

## 3. Results

### 3.1. Arg_11_ Slows Ca^2+^ Transient Decline in Cardiomyocytes

We first screened the effects of the commonly used CPPs Arg_9_, Arg_11_, and TAT (Figure 1A) on Ca^2+^ handling in rat cardiomyocytes. Following 1 h of peptide incubation, field-stimulated Ca^2+^ transients were measured during 1 Hz stimulation at 37 °C. Representative recordings are presented in Figure 1B. In comparison with untreated control cells (Ctr), none of the three CPPs altered the magnitude of Ca^2+^ transients (Figure 1C). However, Arg_11_ treatment induced a slowing of Ca^2+^ transient decay, as indicated by a 14% increase in the tau value for Ca^2+^ removal (Figure 1D; *p* = 0.021). No such effects were noted in cells incubated with TAT or Arg_9_ (Figure 1D).

### 3.2. Polyarginine CPPs Inhibit SR Ca^2+^ Uptake in Ventricular Vesicles, in the Presence and Absence of Attached Cargo

We further investigated the source of the apparent off-target effects of Arg_11_ by examining Ca^2+^ homeostasis in SR vesicles isolated from mouse left ventricles. The vesicles were incubated with each of the CPPs for 5 min at 37 °C, before oxalate-supported Ca^2+^ uptake was initiated by the addition of ATP. Representative recordings and mean measurements are shown in Figure 2B (left and second from left panels, respectively). In agreement with the findings from cardiomyocytes, we observed a marked slowing of Ca^2+^ uptake in vesicles incubated with Arg_11_ in comparison with untreated controls. Interestingly, Arg_9_ also slowed Ca^2+^ uptake, albeit to a lesser extent than Arg_11_, while TAT had no significant effect.

Following Ca^2+^ uptake measurements, thapsigargin was applied to assess the rate of Ca^2+^ leak from the vesicles (Figure 2B, left panel). The ryanodine receptor opener CmC was then applied to measure the vesicular Ca^2+^ content (i.e., the releasable [Ca^2+^]), and these values were used to normalize Ca^2+^ leak measurements. No differences in Ca^2+^ leak were noted between the treatment groups (Figure 2B, right panel), supporting that the observed slowing of Ca^2+^ uptake in the presence of the polyarginines results from the inhibition of SERCA activity.

We next tested whether the observed off-target effects would be of consequence when the polyarginine CPPs were used to deliver cargo. For these experiments, the CPPs were attached to a scrambled cargo control sequence (Figure 2A) and then incubated with the vesicles as previously. Treatment with Arg_11_-cargo continued to significantly slow the Ca^2+^ reuptake rate in comparison with the untreated control experiments (Figure 2C). A similar, but non-significant, tendency was observed in the experiments with Arg_9_-cargo (*p* = 0.129). Again, we did not observe effects of the peptides on Ca^2+^ leak rate (Figure 2C, right panel), supporting an off-target, inhibitory effect on SERCA activity.

### 3.3. SERCA Inhibition by Polyarginine CPPs Is Not Dependent on Phospholamban

In cardiomyocytes, SERCA activity is tightly regulated by its endogenous inhibitor phospholamban (PLB) [22]. We tested whether PLB is required for the observed inhibition of SERCA activity by polyarginine CPPs by examining Ca^2+^ handling in ventricular vesicles from PLB knockout mice [8]. As expected, the vesicles from PLB knockouts exhibited much faster Ca^2+^ uptake overall than the vesicles from wild-type hearts (compare Figure 3A with Figure 2B; *p* < 0.001). However, incubation with the three peptides had similar effects as in wild-type experiments, with Arg_11_ tending to cause the most marked inhibition of Ca^2+^ uptake, both in the absence (Figure 3A) and presence (Figure 3B) of attached cargo. As in the experiments with wild-type vesicles, the peptides did not alter the rate of Ca^2+^ leak from the vesicles in the PLB knockout, supporting that the inhibition of SERCA activity occurs via a mechanism that is independent of PLB.

### 3.4. Polyarginine CPPs Inhibit Ca^2+^ Uptake in SR Vesicles from Rat Soleus, but Not EDL

We next investigated whether the polyarginine CPPs also inhibit SERCA in skeletal muscle, using SR vesicles prepared from rat soleus and EDL. Of note, in the Wistar rat strain, the soleus contains more than 90% slow-twitch (type I) fibres [23] and, like cardiomyocytes, predominantly expresses SERCA2a [24]. Representative recordings and mean measurements of the Ca^2+^ uptake in soleus homogenates (Figure 4A) show that incubation with the test CPPs had similar effects on the Ca^2+^ uptake as observed in the cardiac ventricle. Indeed, while TAT did not alter the rate of Ca^2+^ uptake, both polyarginine CPPs significantly slowed the Ca^2+^ reuptake, with Arg_11_ exhibiting the most marked effect (Figure 4A). Similar inhibition of Ca^2+^ reuptake was observed when cargo was attached to Arg_9_ and Arg_11_ (Figure 4C). We did not observe marked effects of the peptide treatments on SR Ca^2+^ leak or releasable Ca^2+^ (Appendix A).

Strikingly different observations were made using homogenates of the fast-twitch (type II) EDL muscle, which expresses SERCA1 [24] (Figure 4B). While the baseline Ca^2+^ uptake was observed to be markedly faster in the EDL than the soleus (0.077 s^−1^ vs. 0.026 s^−1^, *p* < 0.001), none of the tested peptides altered the rate of Ca^2+^ uptake in the EDL (Figure 4B,D), Ca^2+^ leak, or releasable Ca^2+^ (Appendix A). These findings suggest that the off-target effects of polyarginine CPPs may be specific for the SERCA2 isoforms.

### 3.5. Polyarginine CPPs Inhibit SERCA2 Isoforms, but Not SERCA1 or SERCA3

To further investigate the effect of the Arg-containing CPPs on distinct SERCA isoforms, we employed HEK cells transfected with *SERCA1*, *SERCA2a*, *SERCA2b*, or *SERCA3* constructs. After growing to confluence, the cells were harvested and vesicles were isolated as for the heart and skeletal muscle tissue. Ca^2+^ uptake measurements were then performed with Arg_9_ or Arg_11_ present in the incubation buffer. Neither Arg_9_ nor Arg_11_ altered the Ca^2+^ uptake rate in cells transfected with *SERCA1* or *SERCA3* (Figure 5A,B). However, incubation with Arg_9_ or Arg_11_ significantly slowed Ca^2+^ uptake in cells transfected with *SERCA2a* or *SERCA2b* (Figure 5C,D). As in previous experiments examining cardiac and skeletal muscle homogenates, inhibition by Arg_11_ tended to be more marked than that induced by Arg_9_. These results support that polyarginine CPPs have an off-target inhibitory effect on the SERCA2 isoforms.

### 3.6. Analysis of CPP Binding to SERCA2

Finally, we assessed whether the investigated CPPs directly bind to SERCA2. Using an ELISA-based technique, we tested biotin-labelled control peptides (biotin-cargo_1–9,_ biotin-cargo_8–18_, and biotin-cargo_1–11_) or CPPs (biotin-TAT, biotin-Arg_9_, or biotin-Arg_11_). The interactions were measured with PBS (Figure 6A) or a recombinant SERCA2_111–253_ polypeptide containing the A-domain of SERCA2 [10] coated in the wells (Figure 6B). The ELISA analysis showed negligible binding between any of the biotin-labelled peptides and PBS, and biotin-TAT exhibited similar binding to 12xHis-SERCA2_111–253_ as the control peptides. However, biotin-Arg_9_ binding tended to be higher than the control peptides, and Arg_11_ showed significant binding to SERCA2_111–253_ (Figure 6B). Due to the expected “stickiness” of the tested CPPs with diverse cytosolic protein residues, we additionally tested their binding with another key regulator of cytosolic Ca^2+^ homeostasis, Ca^2+^-calmodulin-dependent protein kinase II (CaMKII) (Figure 6C). We observed significant binding of all three CPPs to a recombinant CaMKII_1-165_ peptide of similar size as that employed in the SERCA2 ELISA experiments. However, we found no preferential binding of Arg_11_ in comparison with TAT or Arg_9_ (Figure 6C).

We further investigated the interaction between Arg_11_ and SERCA2 using surface plasmon resonance to assess the affinity and kinetics of Arg_11_ binding (Figure 7). A biotinylated Arg_11_ peptide was immobilized on an SA chip and exposed to increasing concentrations of 12xHis-SERCA2_111–253_. Strong binding and a slow rate of dissociation were noted, as reflected by a dissociation equilibrium constant (K_D_) of 0.4 ± 0.1 nM, an association rate constant (k_a_) of 2.1 ± 0.7 × 10^5^ M^−1^ s^−1^, and a dissociation rate constant (k_d_) of 6.6 ± 0.9 × 10^−5^ s^−1^ (Figure 7). Taken together with our earlier results, these findings are consistent with inhibition of SERCA2 by Arg_11_ via a direct interaction with the A-domain of the protein.

## 4. Discussion

Our present results indicate that commonly used polyarginine CPPs can have off-target effects on Ca^2+^ homeostasis due to the inhibition of SERCA2. Indeed, our investigations in cardiac, left ventricular myocytes, and homogenates revealed marked inhibition of Ca^2+^ recycling into the SR (Figure 1 and Figure 2B), and similar effects were observed in homogenates from the soleus muscles (Figure 4A). In both tissues, incubation with polyarginines did not alter the rate of SR Ca^2+^ leak, indicating that protracted Ca^2+^ removal from the cytosol stemmed from SERCA inhibition. Importantly, cardiac and soleus muscle both predominantly express the SERCA2a isoform of the Ca^2+^ ATPase [24], and we did not observe altered SR Ca^2+^ uptake in the SERCA1-expressing EDL muscle following treatment with the polyarginine CPPs (Figure 4B). Furthermore, in HEK cell experiments, inhibition of Ca^2+^ uptake was only observed in cells overexpressing SERCA2a or SERCA2b, but not SERCA1 or SERCA3 (Figure 5). Thus, our data support that polyarginine CPPs specifically inhibit the SERCA2 isoforms of this Ca^2+^ ATPase. We observed that this inhibition was most marked following incubation with Arg_11_, although lesser but statistically significant inhibition by Arg_9_ was noted in several experiments (Figure 2B, Figure 3A, Figure 4A, and Figure 5C,D).

How do the polyarginine CPPs interact with SERCA2? Our experiments in myocardial preparations from PLB-knockout mice indicate that this off-target action does not arise via effects on PLB, SERCA’s endogenous inhibitor (Figure 3A). Furthermore, marked SERCA2 inhibition by the polyarginine CPPs was observed in HEK cell experiments, where PLB is also absent (Figure 5). Our ELISA-based analyses instead suggest that Arg_11_, and to a lesser extent Arg_9_, directly bind to the actuator, or A-domain, of SERCA2; an action that was not observed for TAT or control peptides (Figure 6B). Surface plasmon resonance experiments confirmed strong binding between Arg_11_ and this region of the SERCA2 protein, as well as slow dissociation between the two species. The precise nature of the mechanism by which polyarginines bind to the A-domain is unclear, as this residue can interact with various amino acids through different types of interactions. Since arginine is positively charged due to its guanidinium group, it can form ionic interactions (salt bridges) with negatively charged amino acids, such as aspartate and glutamate. While these residues are distributed throughout the A-domain of SERCA2, we note that E238 is present in SERCA2 but not SERCA1 or 3, and that this site is conserved across the three species we have currently examined (human, rat, and mouse). Arginine can also participate in hydrogen bonding with amino acids that contain hydrogen bond donor and acceptor groups, such as serine, threonine, asparagine, and glutamine. The threonine residue T166 is thus of particular interest to our current findings, as it is exclusively expressed within the A-domain of the SERCA2 isoforms, and also conserved across humans, rats, and mice. Finally, the guanidinium group of arginine can form pi-cation interactions with aromatic amino acids like phenylalanine, tyrosine, and tryptophan [25], although these residues are not exclusively expressed at any sites within the A-domain of SERCA2a and 2b. It is interesting to note that all three types of interactions (salt bridges, hydrogen bonds, and pi-cation interactions) are made possible by the high polarity of arginine’s guanidinium group. Thus, while the high polarity of arginine afforded by its guanidinium group allows polyarginine CPPs to interact with phospholipid phosphate groups in the cell membrane, thereby promoting cellular entry [5,6], these same properties of polyarginines are likely tied to their off-target effects.

How does polyarginine binding to SERCA2′s A-domain inhibit the pump’s activity? Importantly, this domain serves as SERCA2′s transduction element, critically coupling ATP hydrolysis with Ca^2+^ transport. The A-domain is physically separated from the nucleotide-binding (N) and phosphorylation (P) domains while in the E1-2Ca state, which may allow polyarginine binding. However, ATPase activity requires that the three domains come together [26]. We postulate that the presence of bound polyarginine peptides may physically inhibit the cojoining of the three functional domains, thereby inhibiting SERCA2 pump function. This hypothesis will be examined in future work.

While it is perhaps somewhat surprising that the off-target inhibition of SERCA2 has been overlooked in previous work employing polyarginine CPPs, we note that these studies have often only compared the CPP-mediated delivery of cargo and scrambled cargo. Our results stress the need to conduct additional control experiments that include an untreated group. An experimental group treated with the CPP of interest alone (without cargo) should also be considered, since scrambled cargo sequences can have unexpected physiological effects. In our experiments, we observed that Arg-homopolymers inhibited SERCA2 activity both in the absence and presence of scrambled cargo, in comparison with untreated controls (Figure 2, Figure 3 and Figure 4). Based on these findings, we urge caution when selecting CPPs for the delivery of bioactive molecules, both in experimental settings and when evaluating CPPs for therapeutic approaches [2,27].

Due to the expression of SERCA2, particularly the SERCA2b isoform, in diverse cell types [24], the widespread consequences of off-target SERCA2 inhibition should be anticipated. Our present work has largely focused on myocytes, and it is well established that alterations in SERCA2 activity in these cells can strongly affect relaxation properties. Indeed, in the heart, lowered cardiomyocyte SERCA2 activity slows myocardial relaxation, causing diastolic dysfunction, a common primary deficit in heart failure [19,28,29,30]. However, lower SERCA2 activity can also contribute to reduced SR Ca^2+^ content and release, i.e., systolic dysfunction in this disease [31]. Importantly, these effects are noted at high physiological pacing frequencies when there is inadequate time for Ca^2+^ resequestration into the SR. Such effects may not occur at slower pacing rates, as we have employed in our cardiomyocyte experiments, where resting the Ca^2+^ levels are returned to steady state values between each stimulation (Figure 1B). In skeletal muscle, slowed SERCA-dependent Ca^2+^ reuptake and slowing of relaxation are an important component of fatigue [32], and the off-target inhibition of SERCA2 may be of particular concern in heart failure patients who exhibit a baseline reduction in SERCA2 activity and susceptibility to skeletal muscle fatigue [33]. Due to its ubiquitous expression pattern, inhibition of SERCA2b is also of potential concern in diverse non-muscle cell types. It is well-known that in Darier’s disease lowered expression/activity of the ATPase due to mutations pathologically affects the skin [34]. However, recent data have also linked this condition to a higher prevalence of severe neuropsychiatric disorders, including schizophrenia and bipolar disorder [35], underscoring the fact that alterations in SERCA2 activity can have broad implications. Here, it should be remembered that off-target inhibition of SERCA2 may not only influence cytosolic Ca^2+^ homeostasis itself, but also the processes that depend on it. For example, SERCA2 activity also influences nuclear Ca^2+^ handling, and thereby transcription [36], as well as mitochondrial homeostasis and metabolism [37,38]. Thus, the downstream effects of SERCA2 manipulation may be difficult to predict. Our observation that the tested CPPs also bound to a recombinant CaMKIIδ_1-165_ fragment (Figure 6C) raises the possibility that these “sticky” peptides have additional detrimental effects on Ca^2+^ homeostasis beyond the scope of the current work.

Based on our observations, it is worth considering alternative non-Arg-based CPPs that may be employed in experimental and therapeutic settings. Importantly, we did not observe inhibition of SERCA2 activity following the treatment of cardiac, soleus, or EDL myocytes with TAT (Figure 1A, Figure 2B, Figure 3A and Figure 4A). In this CPP, only 6 of 9 residues are Arg, in contrast to the chain of 9 or 11 Arg residues that constitute Arg_9_ and Arg_11_, respectively (Figure 1A). Thus, TAT-based cargo delivery may be advantageous over Arg-homopolymers, particularly in settings sensitive to the off-target effects on Ca^2+^ homeostasis. For example, in our recent work, we observed that TAT-based cargo delivery can be used to effectively modulate Ca^2+^ homeostasis, as we employed this CPP to deliver therapeutic SERCA2-activating peptides [21,39]. Here, an off-target inhibition of SERCA2 would be expected if an Arg-homopolymer had been used, opposing the intended actions of the delivered cargo. Other alternative CPPs which may be considered include the polylysines, which exhibit similar transduction efficiency as the polyarginines [40]; protein transduction domains (PTDs) such as Antp (RQIKIWFQNRRMKWKKREF, [41]) and PTD-5 (RRQRRTSKLMKR, [40]); or the amino acid ornithin [42]. In some applications, such as in tumour cells with low pH, histidine can also function as a CPP, as it becomes protonated in low pH settings [43]. Exciting developments in the field of cell-specific CPPs also raise the possibility of minimizing the off-target effects in in situ and in vivo settings. Importantly, these CPPs may bypass liver and kidney metabolism, thus allowing a much lower dose of the CPP to be administered to reach an adequate level in the cell type of interest [44].

In conclusion, our present data show that polyarginine CPPs have previously unrealized off-target effects, as these peptides robustly bind and inhibit SERCA2. Given that SERCA2 is expressed in a broad range of cell-types, and that the manipulation of Ca^2+^ homeostasis can have profound effects on cellular physiology, we urge caution when employing polyarginine CPPs in both experimental and therapeutic settings.

## Figures and Tables

**Figure 1 cells-12-02358-f001:**
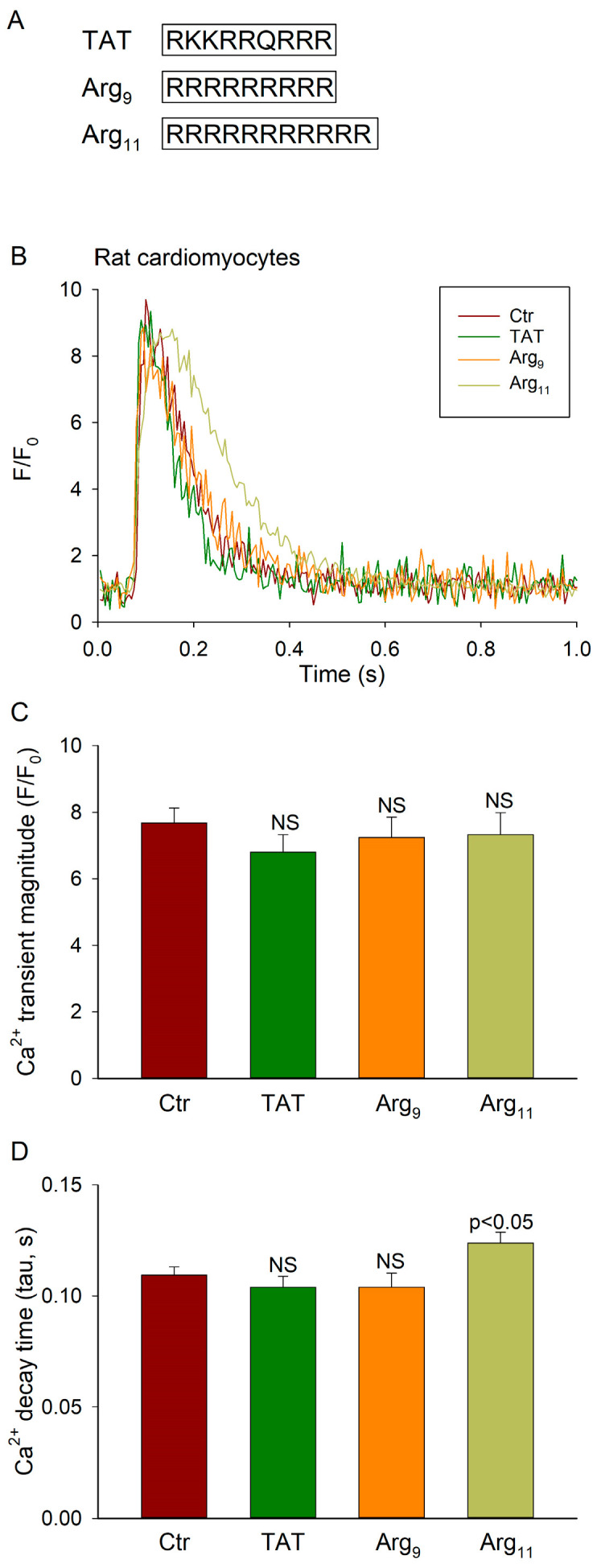
Effect of various CPPs on cardiomyocyte Ca^2+^ transients. (**A**) Three CPPs were investigated with the indicated amino acid sequences. (**B**) Representative field-stimulated Ca^2+^ transients recorded in isolated rat cardiomyocytes incubated with TAT, Arg_9_, or Arg_11_, compared with untreated control cells (Ctr). (**C**) Mean Ca^2+^ transient amplitude. (**D**) Mean measurements of Ca^2+^ transient decay time (tau values). Data are presented as mean ± SEM. Statistical analyses were performed by one-way ANOVA with Tukey’s post hoc correction. *p* values are indicated versus Ctr. NS = not statistically significant. n_cells_ in Ctr, TAT, Arg_9_, Arg_11_ = 37, 35, 39, 31 from 4 hearts in each group.

**Figure 2 cells-12-02358-f002:**
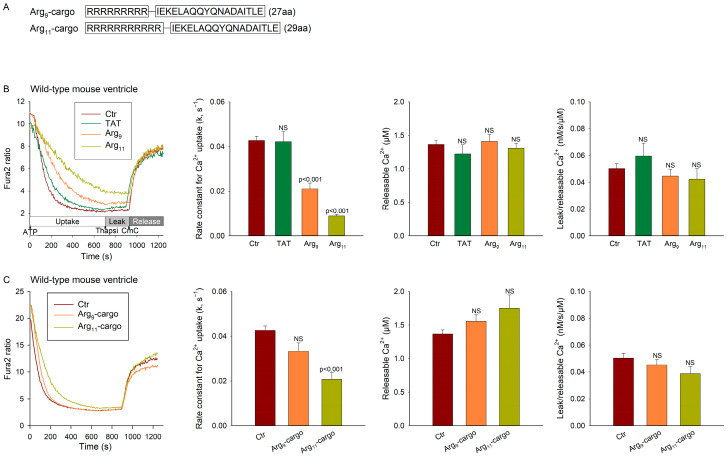
Polyarginine CPPs inhibit Ca^2+^ uptake in SR vesicles isolated from mouse ventricles. (**A**) Ca^2+^ homeostasis was examined in untreated mouse ventricular homogenates (Ctr) compared with homogenates incubated with TAT, Arg_9_, or Arg_11_ in the presence or absence of the indicated, scrambled cargo. (**B**) Oxalate-supported Ca^2+^ uptake was assessed by monitoring the Fura-2 salt fluorescence ratio (340/390) in the test buffer upon addition of Na_4_ATP. SR Ca^2+^ leak was measured as the increase in slope upon the addition of thapsigargin (Thapsi). Application of the ryanodine receptor opener CmC was employed to assess SR Ca^2+^ content, i.e., the releasable [Ca^2+^], and these values were used to normalize leak measurements. Representative Ca^2+^ recordings in the absence of cargo are presented in the left panel, with mean measurements presented in the remaining panels. (**C**) Equivalent data obtained in mouse ventricles incubated with Arg_9_-cargo or Arg_11_-cargo. Data are presented as mean ± SEM. Statistical analyses were performed by one-way ANOVA with Tukey’s post hoc correction. *p* values are indicated versus Ctr. NS = not statistically significant. n_runs_ in Ctr, TAT, Arg_9_, Arg_11_ = 69, 14, 23, 24; n_runs_ in Ctr, Arg_9_-cargo, Arg_11_-cargo = 60, 10, 10 in homogenates from 9 hearts.

**Figure 3 cells-12-02358-f003:**
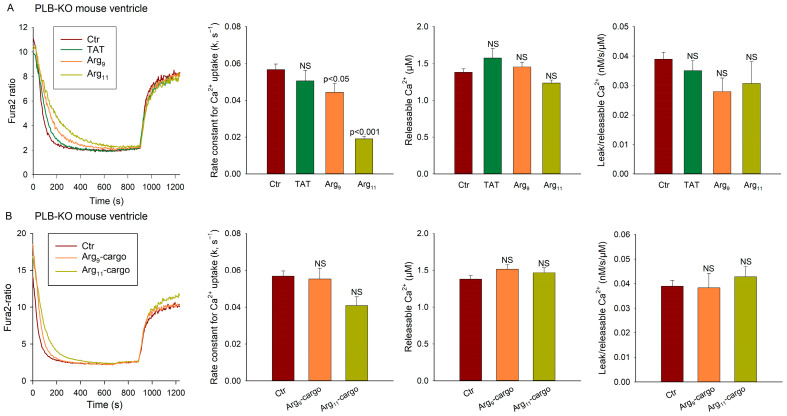
SERCA inhibition by polyarginine CPPs is not dependent on phospholamban. Oxalate-supported Ca^2+^ uptake and SR Ca^2+^ leak were examined in homogenates isolated from the left ventricles of PLB knockout (KO) mice. Comparison was made between untreated homogenates (Ctr) and homogenates incubated with TAT, Arg_9_, or Arg_11_ in the absence (**A**) or presence (**B**) of attached, scrambled cargo. Ca^2+^ uptake, leak, and releasable Ca^2+^ were assessed, as described in Figure 2 (see Methods). Data are presented as mean ± SEM. Statistical analyses were performed by one-way ANOVA with Tukey’s post hoc correction. *p* values are indicated versus Ctr. NS = not statistically significant. n_runs_ in Ctr, TAT, Arg_9_, Arg_11_ = 58, 12, 17, 15; n_runs_ in Ctr, Arg_9_-cargo, Arg_11_-cargo = 58, 9, 10 in homogenates from 9 hearts.

**Figure 4 cells-12-02358-f004:**
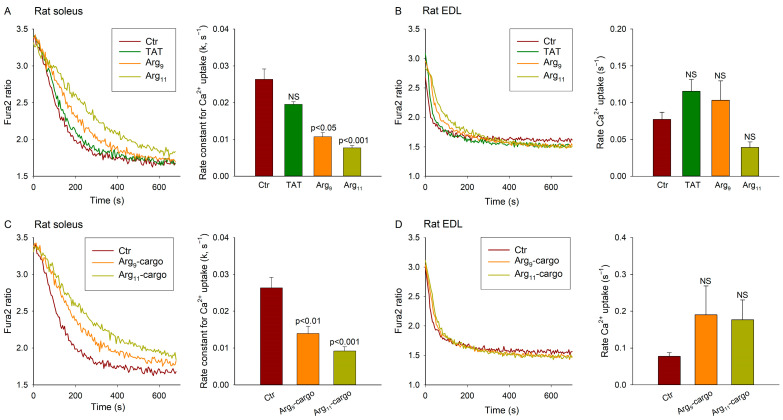
Effect of CPPs on Ca^2+^ homeostasis in skeletal muscle. Oxalate-supported Ca^2+^ uptake was examined in homogenates isolated from the rat soleus and EDL muscles. Comparison was made between untreated homogenates (Ctr) and homogenates incubated with TAT, Arg_9_, or Arg_11_. Arg-containing CPPs significantly inhibited Ca^2+^ uptake in soleus, both in the absence (**A**) or presence (**C**) of attached, scrambled cargo. Peptide incubation did not significantly alter Ca^2+^ uptake in EDL homogenates (**B**,**D**). Data are presented as mean ± SEM. Statistical analyses were performed by one-way ANOVA with Tukey’s post hoc correction. *p* values are indicated versus Ctr. NS = not statistically significant. N_runs_ in Ctr, TAT, Arg_9_, Arg_11_ = 6, 5, 6, 6; n_runs_ in Ctr, Arg_9_-cargo, Arg_11_-cargo = 6, 4, 4 in homogenates from 4 soleus muscles; 7, 5, 6, 5 and 7, 4, 4 in homogenates from 4 EDL muscles.

**Figure 5 cells-12-02358-f005:**
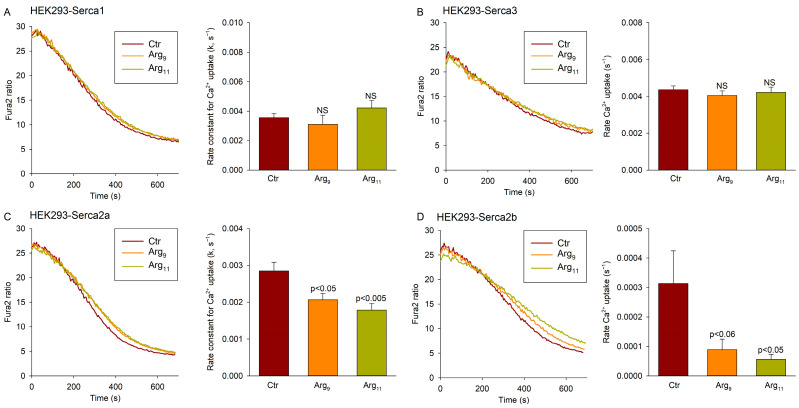
Polyarginine CPPs inhibit SERCA2 isoforms. Oxalate-supported Ca^2+^ uptake was assessed in homogenates of HEK293 cells transfected with *SERCA1* (**A**), *SERCA3* (**B**), *SERCA2a* (**C**), or *SERCA2b* (**D**). Effects of Arg_9_ or Arg_11_ incubation were compared with untreated homogenates. Data are presented as mean ± SEM. Statistical analyses were performed by one-way ANOVA with Tukey’s post hoc correction. *p* values are indicated versus Ctr. NS = not statistically significant. N_runs_ in Ctr, Arg_9_, Arg_11_ = 4, 5, 5 for SERCA1 transfected cells; 4, 5, 5 for SERCA3; 12, 16, 16 for SERCA2a; and 12, 12, 12 for SERCA2b. Homogenates were made from 5–15 dishes of transfected cells in each case.

**Figure 6 cells-12-02358-f006:**
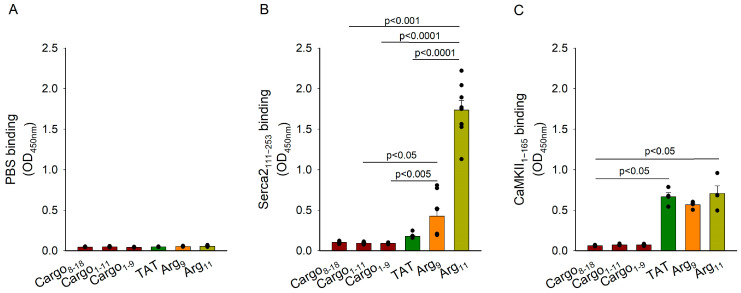
ELISA-based assessment of CPP binding to PBS, SERCA2, and CaMKIIδ. Biotin-labelled control peptides (Cargo_8–18_, Cargo_1–11_, Cargo_1–9_) or CPPs (TAT, Arg_9_, or Arg_11_) were tested for their interaction with PBS (**A**), the 111–253 amino acid region of SERCA2 (**B**), and the 1–165 amino acid region of CaMKIIδ (**C**). Data are presented as mean ± SEM. Statistical differences from control peptide values were performed by a Kruskal–Wallis test with Dunn’s post hoc correction for multiple comparisons. *p* values are indicated for significant differences. *n* = 4–8 runs per peptide (2 × 4 wells per peptide on each plate).

**Figure 7 cells-12-02358-f007:**
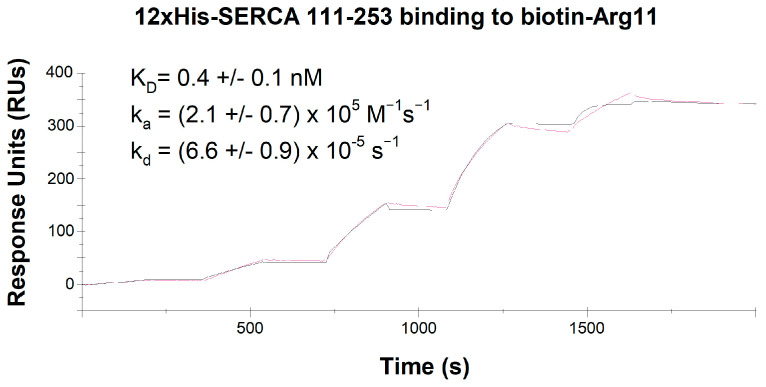
Surface plasmon resonance analysis SERCA2 binding to Arg_11_. Biotin-Arg_11_ was immobilized on SA chips and exposed to recombinant 12xHis-SERCA2_111–253_ protein with stepwise increases in concentration. For the illustrated example, the employed concentrations ranged from 0.7–200 nM. Experimental data are displayed in pink, with fitting by the Langmuir mathematical model superimposed in black. *n* = 4.

## Data Availability

The data presented in this study are available on request from the corresponding author.

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
