# Peer review of "Polyarginine Cell-Penetrating Peptides Bind and Inhibit SERCA2"

_cells, 2023, doi:10.3390/cells12192358_

Round 1

Reviewer 1 Report

This is an interesting manuscript describing experiments showing that cell permeant peptides have the nonspecific action to inhibit Ca pumps of the SERCA2 family in skeletal muscle and cardiomyocytes. The authors use a variety of preparations, including intact cells as well as cell homogenates to look at Ca handling and conclude that these peptides appear to bind and inhibit calcium influx via SERCA2 but not other SERCAs. Overall, the data generally are convincing, although I have some issues that need to be addressed.

The main question is why inhibiting SERCA-mediated Ca accumulation by the ER was reported here to reduce the rate of loss of Ca from the cells, as monitored with fura-2, with no effect seen on transient amplitude? One would suspect in fact that if you inhibit Ca uptake into the ER the cytosolic free Ca transients should be larger in amplitude and fall more quickly due to reduced sequestration. This suggests to me that there may be other effects of the permeant peptides besides SERCA blockade?

Other than this point, the paper appears sound and at the very least introduces a note of caution in the interpretation of experiments carried out using peptides of this type to gain entry to drugs etc conjugated to the peptide.

Reviewer 2 Report

This study describes the effects of cell penetrating peptides (CPP) on the movement of Ca2+ into cellular stores. The authors find that the decay time of Ca2+ concentrations are slowed in the presence of Arg-11 CPP and to a lesser extent Arg-9 CPP but not in the presence of TAT. The same phenomenon appears to be present in isolated rat cardiomyocytes, and homogenates of mouse heart or rat muscle.

Interestingly, the effect is absent in rat EDL, which is consisting more of fast-twitch muscle fibers that express SERCA1 and SERCA3 in preference to SERCA2. The authors suggest that the effects of Arg-11 and Arg-9 on Ca2+ signal decay may be mediated by a direct interaction with SERCA2. They use ELISA and surface plasmon resonance to confirm a physical interaction between SERCA2 and the CPP. In ELISA, the magnitude of interaction is proportional to the CPP’s effect on Ca2+ signal decay.

This is a well-written article with clearly described methodology. The findings are relevant to the understanding of potential physiological effects of CPP, which may be unwanted if CPP are to be used for delivery of therapy.

For ELISA measurements, there are no negative control targets. For instance, it could be useful to exclude interaction of the CPP with SERCA1 or SERCA3 using this technique, if possible. A further analysis of the downstream effects or physiological relevance of CPP interaction with SERCA2 would also be of interest. Do the authors believe that the interaction has an effect e.g. on gene expression and mitochondrial metabolism in cells or on muscle or heart capacity in animals, and are there ways to assess the magnitude of this effect?

In my version, the images are blurry, so high-resolution images will be needed for the final version. Finally, a minor edit on line 249-250, should probably say compare Fig 3A with Fig 2B.
